# DIVERSE VIDEO GENERATION USING A GAUSSIAN PROCESS TRIGGER

**Gaurav Shrivastava and Abhinav Shrivastava**
University of Maryland, College Park
{gauravsh,abhinav}@cs.umd.edu

## ABSTRACT

Generating future frames given a few context (or past) frames is a challenging task. It requires modeling the temporal coherence of videos and multi-modality in terms of diversity in the potential future states. Current variational approaches for video generation tend to marginalize over multi-modal future outcomes. Instead, we propose to explicitly model the multi-modality in the future outcomes and leverage it to sample diverse futures. Our approach, Diverse Video Generator, uses a Gaussian Process (GP) to learn priors on future states given the past and maintains a probability distribution over possible futures given a particular sample. In addition, we leverage the changes in this distribution overtime to control the sampling of diverse future states by estimating the end of on-going sequences. That is, we use the variance of GP over the output function space to trigger a change in an action sequence. We achieve state-of-the-art results on diverse future frame generation in terms of reconstruction quality and diversity of the generated sequences. Webpage - http://www.cs.umd.edu/~gauravsh/dvg.html

## 1 INTRODUCTION

Humans are often able to imagine multiple possible ways that the scene can change over time. Modeling and generating diverse futures is an incredibly challenging problem. The challenge stems from the inherent multi-modality of the task, *i.e.*, given a sequence of past frames, there can be multiple possible outcomes of the future frames. For example, given the image of a "person holding a cup" in Figure. 1, most would predict that the next few frames correspond to either the action "drinking from the cup" or "keeping the cup on the table." This challenge is exacerbated by the lack of real training data with diverse outputs – all real-world training videos come with a single real future and no "other" potential futures. Similar looking past frames can have completely different futures (*e.g.*, Figure. 1). In the absence of any priors or explicit supervision, the current methods struggle with modeling this diversity. Given similar looking past frames, with different futures in the training data, variational methods, which commonly utilize (Kingma & Welling, 2013), tend to average the results to better match to *all* different futures (Denton & Fergus, 2018; Babaeizadeh et al., 2017; Gao et al., 2018; Lee et al., 2018; Oliu et al., 2017). We hypothesize that explicit modeling of future diversity is essential for high-quality, diverse future frame generation.

In this paper, we model the diversity of the future states, given past context, using Gaussian Processes ($\mathcal{GP}$) (Rasmussen, 2006), which have several desirable properties. They learn a prior on potential future given past context, in a Bayesian formulation. This allows us to update the distribution of possible futures as more context frames are provided as evidence and maintain a list of potential futures (underlying *functions* in $\mathcal{GP}$). Finally, our formulation provides an interesting property that is crucial to generating future frames – the ability to estimate *when* to generate a diverse output *vs.* *continue* an on-going action, and a way to *control* the predicted futures.

In particular, we utilize the variance of the $\mathcal{GP}$ at any specific time step as an indicator of whether an action sequence is on-going or finished. An illustration of this mechanism is presented in Figure. 2. When we observe a frame (say at time t) that can have several possible futures, the variance of the $\mathcal{GP}$ model is high (Figure. 2 (left)). Different functions represent potential action sequences that can be generated, starting from this particular frame. Once we select the next frame (at t+2), the $\mathcal{GP}$ variance of the future states is relatively low (Figure. 2 (center)), indicating that an action sequence

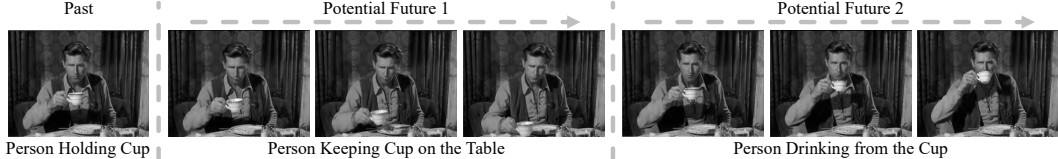

Figure 1: Given "person holding cup," humans can often predict multiple possible futures (*e.g.*,"drinking from the cup" or "keeping the cup on the table.").

is on-going, and the model should continue it as opposed to trying to sample a diverse sample. After the completion of the on-going sequence, the $\mathcal{GP}$ variance over potential future states becomes high again. This implies that we can continue this action (*i.e.*, pick the mean function represented by the black line in Figure. 2 (center)) or try and sample a potentially diverse sample (*i.e.*, one of the functions that contributes to high-variance). This illustrates how we can use $\mathcal{GP}$ to decide when to trigger diverse actions. An example of using $\mathcal{GP}$ trigger is shown in Figure. 2 (right), where after every few frames, we trigger a different action.

Now that we have a good way to model diversity, the next step is to generate future frames. Even after tremendous advances in the field of generative models for image synthesis (Denton & Fergus, 2018; Babaeizadeh et al., 2017; Lee et al., 2018; Vondrick & Torralba, 2017; Lu et al., 2017; Vondrick et al., 2016; Saito et al., 2017; Tulyakov et al., 2018; Hu & Wang, 2019), the task of generating future frames (not necessarily diverse) conditioned on past frames is still hard. As opposed to independent images, the future frames need to obey potential video dynamics that might be on-going in the past frames, follow world knowledge (*e.g.*, how humans and objects interact), *etc.*. We utilize a fairly straightforward process to generate future frames, which utilizes two modules: a frame auto-encoder and a dynamics encoder. The frame auto-encoder learns to encode a frame in a latent representation and utilizes it to generate the original frame back. The dynamics encoder learns to model dynamics between past and future frames. We learn two independent dynamics encoders: an LSTM encoder, utilized to model on-going actions and the $\mathcal{GP}$ encoder (similar to (Srivastava et al., 2015)), and a $\mathcal{GP}$ encoder, utilized to model transitions to new actions. The variance of this $\mathcal{GP}$ encoder can be used as a trigger to decide when to sample new actions. We train this framework end-to-end. We first provide an overview of $\mathcal{GP}$ formulation and scalable training techniques in §3, and then describe our approach in §4.

Comprehensively evaluating diverse future frames generation is still an open research problem. Following recent state-of-the-art, we will evaluate different aspects of the approach independently. The quality of generated frames is quantified using image synthesis/reconstruction per-frame metrics: SSIM (Wang et al., 2019; Sampat et al., 2009), PSNR, and LPIPS (Zhang et al., 2018; Dosovitskiy & Brox, 2016; Johnson et al., 2016). The temporal coherence and quality of a short video clip (16 neighboring frames) are jointly evaluated using the FVD (Unterthiner et al., 2018) metric. However, high-quality, temporarily coherent frame synthesis does not evaluate diversity in predicted frames. Therefore, to evaluate diversity, since there are no multiple ground-truth futures, we propose an alternative evaluation strategy, inspired by (Villegas et al., 2017b): utilizing action classifiers to evaluate whether an *action switch* has occurred or not. A change in action indicates that the method was able to sample a diverse future. Together, these metrics can evaluate if an approach can generate multiple high-quality frames that temporally coherent and diverse. Details of these metrics and baselines, and extensive quantitative and qualitative results are provided in §5.

To summarize, our contributions are: (a) modeling the diversity of future states using a $\mathcal{GP}$, which maintains priors on future states given the past frames using a Bayesian formulation (b) leveraging the changing $\mathcal{GP}$ distribution over time (given new observed evidence) to estimate when an on-going action sequence completes and using $\mathcal{GP}$ variance to *control* the triggering of a diverse future state. This results in state-of-the-art results on future frame generation. We also quantify the diversity of the generated sequences using action classifiers as a proxy metric.

## 2 RELATED WORK

Understanding and predicting the future, given the observed past, is a fundamental problem in video understanding. The future states are inherently multi-modal and capturing their diversity finds direct applications in many safety-critical applications (*e.g.*, autonomous vehicles), where it is critical

Figure 2: An illustration of using $\mathcal{GP}$ variance to control sampling on-going actions *vs.* new actions.

to model different future modes. Earlier techniques for future prediction (Yuen & Torralba, 2010; Walker et al., 2014) relied on searching for matches of past frames in a given dataset and transferring the future states from these matches. However, the predictions were limited to symbolic trajectories or retrieved future frames. Given the modeling capabilities of deep representations, the field of future frame prediction tremendous progress in recent years. One of the first works on video generation (Srivastava et al., 2015) used a multi-layer LSTM network to learn representations of video sequences in a deterministic way. Since then, a wide range of papers (Oliu et al., 2017; Cricri et al., 2016; Villegas et al., 2017a; Elsayed et al., 2019; Villegas et al., 2019; Wang et al., 2019; Castrejón et al., 2019) have built models that try to incorporate stochasticity of future states. Generally, this stochasticity lacks diverse high-level semantic actions.

Recently, several video generation models have utilized generative models, like variational auto-encoders (VAEs) (Kingma & Welling, 2013) and generative adversarial networks (GANs) (Goodfellow et al., 2014), for this task. One of the first works by Xue et al. (2016) utilized a conditional VAE (cVAE) formulation to learn video dynamics. Similar to our approach, their goal was to model the frame prediction problem in a probabilistic way and synthesizing many possible future frames from a single image. Since then, several works have utilized the cVAE for future generation (Babaeizadeh et al., 2017; Denton & Fergus, 2018). The major drawback of using the cVAE approach is that its objective function marginalizes over the multi-modal future, limiting the diversity in the generated samples (Bhattacharyya et al., 2018). GAN-based models are another important class of synthesis models used for future frame prediction or video generation (Vondrick & Torralba, 2017; Lu et al., 2017; Vondrick et al., 2016; Saito et al., 2017; Tulyakov et al., 2018; Hu & Wang, 2019). However, these models are very susceptible to mode collapse (Salimans et al., 2016), *i.e.*, the model outputs only one or a few modes instead of generating a wide range of diverse output. The problem of mode collapse is quite severe for conditional settings, as demonstrated by (Isola et al., 2016; Zhu et al., 2017; Mathieu et al., 2015). This problem is worse in the case of diverse future frame generation due to the inherent multi-modality of the output space and lack of training data.

Another class of popular video generation models is hierarchical prediction (Walker et al., 2017; Villegas et al., 2017b; Wichers et al., 2018; Cai et al., 2018). These models decompose the problems into two steps. They first predict a high-level structure of a video, like a human pose, and then leverage that structure to make predictions at the pixel level. These models generally require additional annotation for the high-level structure for training.

Unlike these approaches, Our approach explicitly focuses on learning the distribution of diverse futures using a $\mathcal{GP}$ prior on the future states using a Bayesian formulation. Moreover, such $\mathcal{GP}$ approaches have been used in the past for modeling the human dynamics as demonstrated by (Wang et al., 2008; Hardy et al., 2014; Moon & Pavlovic, 2008). However, due to the scalability issues in $\mathcal{GP}$, these models were limited to handling low dimensional data, like human pose, lane switching, path planning, etc. To the best of our knowledge, ours is the first approach that can process video sequences to predict when an on-going action sequence completes and control the generation of a diverse state.

## 3 REVIEW OF GAUSSIAN PROCESS

A Gaussian process ($\mathcal{GP}$) (Rasmussen, 2006) is a Bayesian non-parametric approach that learns a joint distribution over functions that are sampled from a multi-variate normal distribution. Consider a data set consisting of n data-points $\{\text{inputs}, \text{targets}\}_1^n$, abbreviated as $\{X, Y\}_1^n$, where the inputs are denoted by $X = \{\mathbf{x}_1, \ldots, \mathbf{x}_n\}$, and targets by $Y = \{\mathbf{y}_1, \ldots, \mathbf{y}_n\}$. The goal of $\mathcal{GP}$ is to learn

an unknown function $f$ that maps elements from input space to a target space. A $\mathcal{GP}$ regression formulates the functional form of $f$ by drawing random variables from a multi-variate normal distribution given by $[f(\mathbf{x}_1), f(\mathbf{x}_2), \ldots, f(\mathbf{x}_n)] \sim \mathcal{N}(\mu, K_{X,X})$, with mean $\mu$, such that $\mu_i = \mu(\mathbf{x}_i)$, and $K_{X,X}$ is a covariance matrix. $(K_{X,X})_{ij} = k(\mathbf{x}_i, \mathbf{x}_j)$, where $k(\cdot)$ is a kernel function of the $\mathcal{GP}$. Assuming the $\mathcal{GP}$ prior on $f(X)$ with some additive Gaussian white noise $\mathbf{y}_i = f(\mathbf{x}_i) + \epsilon$, the conditional distribution at any unseen points $X_*$ is given by:

$$\mathbf{f}_*|X_*, X, Y \sim \mathcal{N}(E[\mathbf{f}_*], \mathrm{Cov}[\mathbf{f}_*]), \text{ where} \tag{1}$$
$$E[\mathbf{f}_*] = \mu_{X_*} + K_{X_*,X}[K_{X,X} + \sigma^2 I]^{-1} Y$$
$$\mathrm{Cov}[\mathbf{f}_*] = K_{X_*,X_*} - K_{X_*,X}[K_{X,X} + \sigma^2 I]^{-1} K_{X,X_*}$$

### 3.1 LEARNING AND MODEL SELECTION

We can derive the marginal likelihood for the $\mathcal{GP}$ by integrating out the $f(x)$ as a function of kernel parameters alone. Its logarithm can be defined analytically as:

$$\log\left(p\left(Y|X\right)\right) = -\frac{1}{2}\left(Y^T \left(K_\theta + \sigma^2 I\right)^{-1} Y + \log\left|K_\theta + \sigma^2 I\right|\right) + \text{const}, \tag{2}$$

where $\theta$ denotes the parameters of the covariance function of kernel $K_{X,X}$. Notice that the marginal likelihood involves a matrix inversion and evaluating a determinant for $n \times n$ matrix. A naïve implementation would require cubic order of computations $\mathcal{O}(n^3)$ and $\mathcal{O}(n^2)$ of storage, which hinders the use of $\mathcal{GP}$ for a large dataset. However, recent researches have tried to ease these constraints under some assumptions.

### 3.2 SCALABLE $\mathcal{GP}$

The model selection and inference of $\mathcal{GP}$ requires a cubic order of computations $\mathcal{O}(n^3)$ and $\mathcal{O}(n^2)$ of storage which hinders the use of $\mathcal{GP}$ for a large dataset. Titsias (2009) proposed a new variational approach for sparse approximation of the standard $\mathcal{GP}$ which jointly infers the inducing inputs and kernel hyperparameters by optimizing a lower bound of the true log marginal likelihood, resulting in $\mathcal{O}(nm^2)$ computation, where $m < n$. Hensman et al. (2013) proposed a new variational formulation of true log marginal likelihood that resulted in a tighter bound. Another advantage of this formulation is that it can be optimized in a stochastic (Hensman et al., 2013) or distributed (Dai et al., 2014; Gal et al., 2014) manner, which is well suited for our frameworks which use stochastic gradient descent. Further, recent works (Wilson & Nickisch, 2015; Wilson et al., 2015; 2016) have improved the scalability by reducing the learning to $\mathcal{O}(n)$ and test prediction to $\mathcal{O}(1)$ under some assumptions.

In this work, for scalability, we leverage the SVGP approach proposed by Hensman et al. (2013). It proposes a tighter bound for the sparse GP introduced by Titsias (2009), which uses pseudo inputs $\mathbf{u}$ to lower bound the log joint probability over targets and pseudo inputs. SVGP introduces a multivariate normal variational distribution $q(\mathbf{u}) = \mathcal{N}(\mathbf{m}, \mathbf{S})$, where the parameters $\mathbf{m}$ and $\mathbf{S}$ are optimized using the evidence lower bound or ELBO (eq. 3) of true marginal likelihood (eq. 2). The pseudo inputs, $\mathbf{u}$, depend on variational parameters $\{\mathbf{z}_m\}_{m=1}^M$, where $M = \dim(\mathbf{u}) \ll N$. Therefore, the ELBO for SVGP is

$$\mathcal{L}_{\text{svgp}}(Y, X) = \mathbb{E}_{q(\mathbf{u})}\left[\log p\left(Y, \mathbf{u}|X, Z\right)\right] + H[q(\mathbf{u})], \tag{3}$$

where the first term was proposed by Titsias (2009), and the modification by Hensman et al. (2013) introduces the second term. For details about the pseudo inputs $\mathbf{u}$ and variational parameters $\mathbf{z}_i$, we refer the readers to (Titsias, 2009; Hensman et al., 2013).

In this work, we build on the sparse $\mathcal{GP}$ approach from GPytorch (Gardner et al., 2018), which implements (Hensman et al., 2013).

## 4 OUR APPROACH

Given a set of observed frames, our goal is to generate a diverse set of future frames. Our model has three modules: (a) a frame auto-encoder (or encoder-generator), (b) an LSTM temporal dynamics

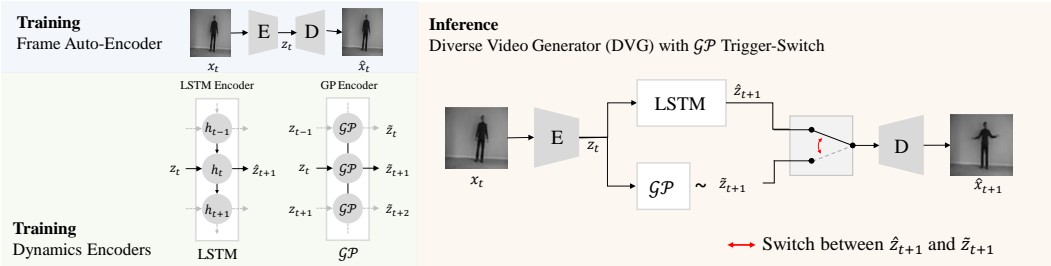

Figure 3: An overview of the proposed Diverse Video Generator (**DVG**).

encoder, and (c) a $\mathcal{GP}$ temporal dynamics encoder to model priors and probabilities over diverse potential future states.

The frame encoder maps the frames onto a latent space, that is later utilized by temporal dynamics encoders and frame generator to synthesize the future frames. For inference, we use all three modules together to generate future frames, and use the $\mathcal{GP}$ as a trigger to switch to diverse future states. Below we describe each module in detail.

## 4.1 FRAME AUTO-ENCODER

The frame encoder network is a convolution encoder which takes frame $\mathbf{x}_t \in \mathbb{R}^{H \times W}$ and maps them to the latent space $\mathbf{z}_t \in \mathbb{R}^d$, where $H \times W$ is the input frame size and $d$ is the dimension of latent space respectively. Therefore, $f_{\text{enc}} : \mathbb{R}^{H \times W} \rightarrow \mathbb{R}^d$, *i.e.*, $\mathbf{z}_t = f_{\text{enc}}(\mathbf{x}_t)$. Similarly, the decoder or generator network, utilizes the latent feature to generate the image. Therefore, $f_{\text{gen}} : \mathbb{R}^d \rightarrow \mathbb{R}^{H \times W}$, *i.e.*, $\hat{\mathbf{x}}_t = f_{\text{gen}}(\mathbf{z}_t)$. We borrow the architectures for both encoder and generator networks from (Denton & Fergus, 2018), where the frame encoder is convolutional layers from VGG16 network (Simonyan & Zisserman, 2015) and the generator is a mirrored version of the encoder with pooling layers replaced with spatial up-sampling, a sigmoid output layer, and skip connections from the encoder network to reconstruct image. $\mathcal{L}_{\text{gen}}(\mathbf{x}_t, \hat{\mathbf{x}}_t) = \|\mathbf{x}_t - \hat{\mathbf{x}}_t\|^2$ is the reconstruction loss for frame auto-encoder. This auto-encoder is illustrated in Figure. 3 (stage 1).

## 4.2 LSTM TEMPORAL DYNAMICS ENCODER

The first dynamics we want to encode is of an on-going action sequence, *i.e.*, if an action sequence is not completed yet, we want to continue generating future frames of the same sequence till it finishes. This module $f_{\text{LSTM}} : \mathbb{R}^d \rightarrow \mathbb{R}^d$, has a fully-connected layer, followed by two LSTM layers with 256 cells each, and a final output fully-connected layer. The output fully-connected layer takes the last hidden state from LSTM ($\mathbf{h}_{t+1}$) and outputs $\hat{\mathbf{z}}_{t+1}$ after $\tanh(\cdot)$ activation. Therefore, $\hat{\mathbf{z}}_{t+1} = f_{\text{LSTM}}(\mathbf{z}_t)$. The training loss is given by $\mathcal{L}_{\text{LSTM}} = \sum_{t=1}^T \|\mathbf{z}_{t+1} - \hat{\mathbf{z}}_{t+1}\|^2$, where $T$ are the total number of frames (both past and future) used for training. This simple dynamics encoder, inspired by (Srivastava et al., 2015) and illustrated in Figure. 3 (stage 2), is effective and performs well on standard metrics.

## 4.3 $\mathcal{GP}$ TEMPORAL DYNAMICS ENCODER

Next, we want to learn the priors for potential future states by modeling the correlation between past and future states using a $\mathcal{GP}$ layer. Given a past state, this temporal dynamics encoder captures the distribution over future states. This enables us to use the predictive variance to decide when to sample diverse outputs, and provides us with a mechanism to sample diverse future states. The input to the $\mathcal{GP}$ layer is $\mathbf{z}_t$ and the output is a mean and variance, which can be used to sample $\tilde{\mathbf{z}}_{t+1}$. The loss function for the $\mathcal{GP}$ dynamics encoder follows from eq. 3, $\mathcal{L}_{\text{GP}} = -\mathcal{L}_{\text{svgp}}(\mathbf{z}_{t+1}, \mathbf{z}_t)$.

Unlike LSTM, the $\mathcal{GP}$ layer does not have hidden or transition states; it only models pair-wise correlations between past and future frames (illustrated in Figure. 3 (stage 2)). In this work, we use the automatic relevance determination (ARD) kernel, denoted by $k(\mathbf{z}, \mathbf{z}') = \sigma_{\text{ARD}}^2 \exp\left(-0.5 \sum_{j=1}^d \omega_j (z_j - z_j')^2\right)$, parameterized by learnable parameters $\sigma_{\text{ARD}}$ and $\{\omega_1, \ldots, \omega_d\}$. This $\mathcal{GP}$ layer is implemented using GPyTorch (Gardner et al., 2018) (refer to §3).

## 4.4 TRAINING OBJECTIVE

All three modules, frame auto-encoder and the LSTM and $\mathcal{GP}$ temporal encoders, are jointly trained using the following objective function:

$$
\mathcal{L}_{\text{DVG}} = \sum_{t=1}^{T} \Big( \overbrace{\lambda_1 \mathcal{L}_{\text{gen}}(\mathbf{x}_t, \hat{\mathbf{x}}_t)}^{\text{Frame Auto-Encoder}} + \overbrace{\lambda_2 \mathcal{L}_{\text{gen}}(\mathbf{x}_t, f_{\text{gen}}(\hat{\mathbf{z}}_t))}^{\text{LSTM Frame Generation}} + \overbrace{\lambda_3 \mathcal{L}_{\text{gen}}(\mathbf{x}_t, f_{\text{gen}}(\tilde{\mathbf{z}}_t))}^{\mathcal{GP} \text{ Frame Generation}} +
$$
$$
\underbrace{\lambda_4 \mathcal{L}_{\text{LSTM}}(\mathbf{z}_{t+1}, \hat{\mathbf{z}}_{t+1})}_{\text{LSTM Dynamics Encoder}} + \underbrace{\lambda_5 \mathcal{L}_{\text{GP}}(\mathbf{z}_{t+1}, \mathbf{z}_t)}_{\mathcal{GP} \text{ Dynamics Encoder}} \Big)
\tag{4}
$$

where $[\lambda_1, \ldots, \lambda_5]$ are hyperparameters. There are three frame generation losses, each utilizing different latent code: $\mathbf{z}_t$ from frame encoder, $\hat{\mathbf{z}}_t$ from LSTM encoder, and $\tilde{\mathbf{z}}_t$ from $\mathcal{GP}$ encoder. In additional, there are two dynamics encoder losses, one each for LSTM and $\mathcal{GP}$ modules. Empirically, we observed that the model trains better with higher values for $\lambda_1, \lambda_2, \lambda_4$, possibly because $\mathcal{GP}$ is only used to sample a diverse state and all other states are sampled from the LSTM encoder.

## 4.5 INFERENCE MODEL OF DIVERSE VIDEO GENERATOR (DVG)

During inference, we put together the three modules described above as follows. The output of the frame encoder $\mathbf{z}_t$ is given as input to both LSTM and $\mathcal{GP}$ encoders. The LSTM outputs $\hat{\mathbf{z}}_{t+1}$ and the $\mathcal{GP}$ outputs a mean and a variance. The variance of $\mathcal{GP}$ can be used to decide if we want to continue an on-going action or generate new diverse output, a process we call **trigger switch**. If we decide to stay with the on-going action, LSTM's output $\hat{\mathbf{z}}_{t+1}$ is provided to the decoder to generate the next frame. If we decide to *switch*, we sample $\tilde{\mathbf{z}}_{t+1}$ from the $\mathcal{GP}$ and provide that as input to the decoder. This process is illustrated in Figure. 3 (stage 3). The generated future frame is used as input to the encoder to output the next $\mathbf{z}_{t+1}$; this process is repeated till we want to generate frames.

## 4.6 TRIGGER SWITCH HEURISTICS

An important decision for a diverse future generation is when to continue the current action and when to switch to a new action. We use the $\mathcal{GP}$ to switch to new states. We use two heuristics to decide when to generate diverse actions: a deterministic switch and a $\mathcal{GP}$ trigger switch. For the deterministic switch, we do not use the variance of the $\mathcal{GP}$ as a trigger, and switch every 15 frames. Each switch uses the sample from $\mathcal{GP}$ as the next future state. This enables us to have a consistent sampling strategy across generated samples. For the $\mathcal{GP}$ trigger switch, we compare the current $\mathcal{GP}$ variance with the mean of the variance of the last 10 states. If the current variance is larger than two standard deviations, we trigger a switch. This variable threshold allows the diverse video generator to trigger switches based on evidence, which can vary widely across samples.

## 5 EXPERIMENTS

Next, we describe the experimental setup, datasets we use, qualitative and quantitative results. We evaluate our models on four datasets and compare it with the state-of-the-art baselines. All models use 5 frames as context (past) during training and learn to predict the next 10 frames. However, our model is not limited to generating just 10 frames. All our models are trained using Adam optimizer.

**KTH Action Recognition Dataset.** The KTH action dataset (Schuldt et al., 2004) consists of video sequences of 25 people performing six different actions: walking, jogging, running, boxing, hand-waving, and hand-clapping. The background is uniform, and a single person is performing actions in the foreground. Foreground motion of the person in the frame is fairly regular.

**BAIR pushing Dataset.** The BAIR robot pushing dataset (Ebert et al., 2017) contains the videos of table mounted sawyer robotic arm pushing various objects around. The BAIR dataset consists of different actions given to the robotic arm to perform.

**Human3.6M Dataset.** Human3.6M (Ionescu et al., 2014) dataset consists of 10 subjects performing 15 different actions. We did not use the pose information from the dataset.

**UCF Dataset.** This dataset (Soomro et al., 2012) consists of 13,320 videos belonging to 101 different action classes. We sub-sample a small dataset for qualitative evaluation on complex videos. Our

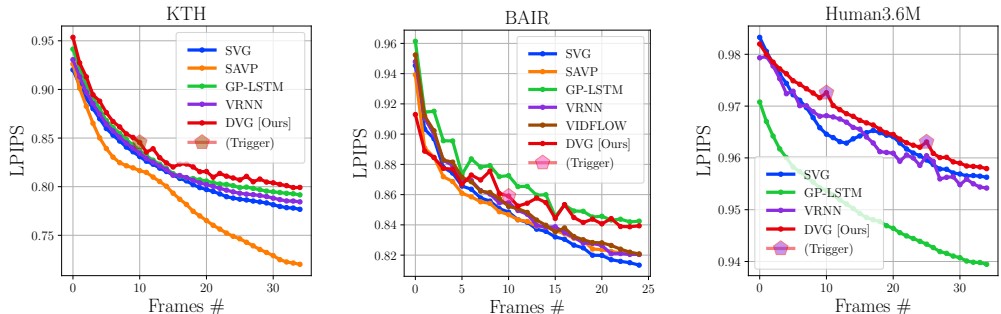

Figure 4: **LPIPS Quantitative Results** on KTH, BAIR, and Human3.6M datasets. All methods use the best matching sample out of 100 random samples. We used fixed trigger heuristic to keep trigger point for each sample the same for our approach.

subset consists of 7 classes: Bench press, Bodyweight squats, Clean and Jerk, Pull-ups, Push-ups, Shotput, Tennis-Swing, Lunges and Fencing. The background of this dataset can bias our diversity evaluation metric. Therefore, we only include this dataset only for qualitative evaluation.

## 5.1 BASELINES

We compared our method with the following prior works. Further, wherever available, we used either the official implementation or the pre-trained models for the baselines uploaded by their respective authors.

**SVG-LP** (Denton & Fergus, 2018): Stochastic Video Generation with a Learned Prior (SVG-LP) is a VAE-based method that uses a latent space prior for generating video sequences. It outperforms other VAE-based approaches (*e.g.*, SV2P (Babaeizadeh et al., 2017)). It also uses an LSTM-based dynamics model. This model similar to ours except that we use a $\mathcal{GP}$ to model the prior on future states to aid with multi-modal outputs.

**SAVP** (Lee et al., 2018):Stochastic Adversarial Video Prediction (SAVP) is a VAE-GAN based approach that combines the best of both families of approaches. It also uses the LSTM dynamics model.

**Conditional VRNN** (Castrejón et al., 2019): Condition variational RNN leverages the flexibility of hierarchical latent variable models to increase the expressiveness of the latent space.

**VidFlow** (Kumar et al., 2019): VideoFlow model uses a normalizing flow approach that enables direct optimization of the data likelihood.

**GP-LSTM**: We train a model inspired by (Al-Shedivat et al., 2016), where the dynamics model is a $\mathcal{GP}$, which uses recurrent kernels modeled by an LSTM. This method is closest to ours since it utilizes the constructs of both $\mathcal{GP}$ and LSTM. However, they train a single dynamics model and have no way to *control* the generation of future states.

We refer to our model as **Diverse Video Generator** (DVG), which uses a $\mathcal{GP}$ trigger switch. We also study heuristic switching at **15** and **35** frames for ablation analysis. We provide additional ablation analysis in the **supplementary material**, for models with RNNs and GRUs instead of LSTM.

## 5.2 METRICS

Evaluation of generated videos in itself is an open research problem with new emerging metrics. In this work, we tried our best to cover all published metrics which have been used for evaluating future frame generation models.

**Accuracy of Reconstruction.** One way to evaluate a video generation model is to check how close the generated frames are to the ground-truth. Since the models are intended to be stochastic or diverse for variety in prediction, this is achieved by sampling a finite number of future sequences from a model and evaluating the similarity between the best matching sequence to the ground-truth and the ground truth sequence. Previous works (Denton & Fergus, 2018; Babaeizadeh et al., 2017; Lee et al., 2018) used traditional image reconstruction metrics, SSIM and PSNR, to measure the

Table 1: **Quantitative results** on KTH, BAIR, Human3.6M datasets. For the **FVD Score**, all methods use the best matching sample out of 100 random samples and lower numbers are better. For the **Diversity Score**, we compute the score across 50 generated samples, for 500 starting sequences, and higher numbers are better.

| Model | Trigger | FVD Score ($\downarrow$) | | | Diversity Score ($\uparrow$) (frames: [10,25]) | | Diversity Score ($\uparrow$) (frames: [25,40]) | |
|---|---|---|---|---|---|---|---|---|
| | | KTH | BAIR | Human3.6M | KTH | Human3.6M | KTH | Human3.6M |
| SVG-LP | - | 156.35 | 270.04 | 718.04 | 20.10 | 4.8 | 21.20 | 4.6 |
| SAVP | - | 65.98 | 126.75 | - | 26.60 | - | 24.50 | - |
| GP-LSTM | - | 92.34 | 197.49 | 604.75 | 31.40 | 5.4 | 30.90 | 6.0 |
| VidFlow | - | - | 124.81 | - | . - | - | - | - |
| VRNN | - | 67.26 | 134.81 | 523.45 | 32.50 | 5.6 | 31.80 | 5.9 |
| DVG [**ours**] | @15,35 | **65.69** | 123.08 | **479.43** | **48.30** | 9.3 | 46.20 | 9.0 |
| DVG [**ours**] | $\mathcal{GP}$ | 69.63 | **120.03** | 496.89 | 47.71 | **10.8** | **48.10** | **10.1** |

similarity between the generated samples and ground-truth. As shown by (Zhang et al., 2018; Dosovitskiy & Brox, 2016; Johnson et al., 2016; Unterthiner et al., 2018), these metrics are not suited for video evaluation because of their susceptibility towards perturbation like blurring, structural distortion, *etc.*. We include these metrics in our **supplementary material** for the sake of completeness. We also evaluate the similarity of our generated sequences using recently proposed perceptual metrics, VGG cosine similarity (LPIPS), and Frechet Video Distance (FVD) score. **LPIPS** or Learned Perceptual Image Patch Similarity is a metric developed to quantify the perceptual distance between two images using deep features. Several works (Zhang et al., 2018; Dosovitskiy & Brox, 2016; Johnson et al., 2016) show that this metric is much more robust to perturbation like distortion, blurriness, warping, color shift, lightness shift, *etc.* **Frechet Video Distance** (FVD score) (Unterthiner et al., 2018) is a deep metric designed to evaluate the generated video sequences. As is standard practice, all methods use the best matching sample out of 100 randomly generated samples.

**Diversity of Sequences.** Reconstruction accuracy of generated samples only implies that there is at least one generated sequence close to the ground-truth. However, these metrics do not capture the inherent multi-modal nature of the task. Aside from being able to generate samples close to ground truth, a video generation model should be able to represent diversity in its generated video sequences. Therefore, we propose a metric inspired by (Villegas et al., 2017b) that utilizes a video classifier to quantify the diversity of generated sequences. The action classifier is trained on the respective datasets (KTH and Humans3.6M). Note that we cannot utilize this metric for the BAIR dataset since we do not have corresponding action labels. For the diversity metric, we use 500 starting sequences of 5 frames, sample 50 future sequences of 40 frames. We ignore the first 5 generated frames since they are likely correlated with the ground-truth and correspond to the ongoing sequences. Then, we evaluate the next two clips, made of frames $[10, 25]$ and $[25, 40]$. Ideally, a method that can generate diverse sequences will generate more diverse clips as time progresses. For the **Diversity Score**, we compute the mean number of generated clips that changed from the on-going action as classified by the classifier. More concretely, if $N$ is the total number of generated clips ($N = 25000$ for us ), $c_i$ is the ground-truth class, $\hat{c}_i$ is the predicted class, and $\mathbb{1}(\cdot)$ is the indicator function if the the parameter is correct, then Diversity Score $= \frac{1}{N} \sum_N \mathbb{1}(c_i \neq \hat{c}_i)$.

## 5.3 RESULTS

### 5.3.1 QUANTITATIVE RESULTS (RECONSTRUCTION)

We report the quantitative results on KTH, BAIR, and Human3.6M datasets using the LPIPS metric in Figure. 4, and FVD metric in Table 1. For comparisons with baselines in Figure. 4, we see that on the KTH and Human3.6M dataset, our approach generally performs on-par or better than the baselines. In fact, except for SAVP, all methods are very close to each other. For the Human3.6M dataset, the GP-LSTM baseline performs poorly, and all other methods are similar, with ours being better than others. On the BAIR dataset, we notice that our GP-LSTM baseline performs better, closely followed by our approach. Again, SAVP performs worse on all metrics. For the FVD metric (Table 1), variants of our approach achieve state-of-the-art results on all datasets. Note that using a fixed trigger at frames **15** and **35** leads to better FVD scores for KTH and Human3.6M dataset,

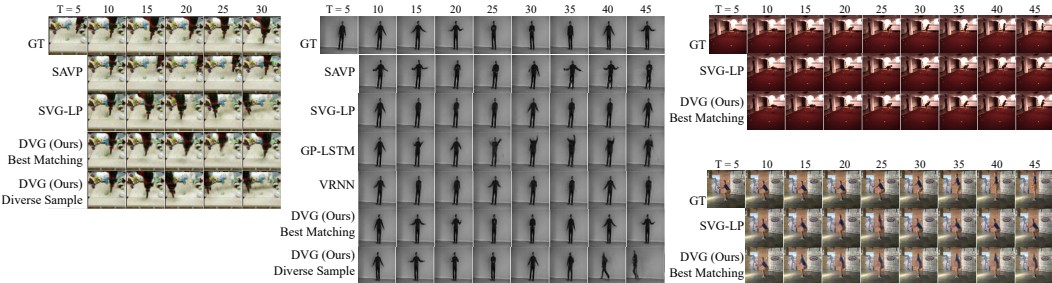

Figure 5: **Qualitative Results** on BAIR (left), KTH (center), Human3.6M (right, top), and UCF (right, bottom) datasets. First row is the ground-truth video in each figure (with the last frame of the provided 5 frames is shown as 'GT'). Every $5^{\text{th}}$ frame is shown.

while $\mathcal{GP}$ trigger performs better for the BAIR dataset. All settings, except one, of our approach perform better than the baselines.

### 5.3.2 QUANTITATIVE RESULTS (DIVERSITY)

We report the quantitative results using the proposed diversity score in Table 1. We notice for the KTH dataset that SVG-LP/SAVP baseline change actions 20.1%/26.6% of the time in the first clip and 21.2%/24.5% of the time in the second clip. In comparison, our approach gives the diversity score of 48.53% and 48.10% for the first and second clips, respectively. As can be observed, the $\mathcal{GP}$ trigger results in considerably higher diversity as the sequence progresses. On the Human3.6M dataset, the difference between the scores of baselines and our methods is $\sim$6%. The overall score drop between the KTH and the Human3.6M datasets on diversity metric can be accounted to actions performed in the videos are very distinct. Besides, cameras are placed far off from the person performing actions making it harder for the models to generate diverse samples.

We further analyze common action triggers for the $\mathcal{GP}$ trigger and notice that it separates the moving (walk, jog, run) and still (clap, wave, box) actions, and common action switches are within each cluster; *e.g.*, walk $\leftrightarrow$ jog, wave $\leftrightarrow$ clap. More analysis is provided in the **supplementary**.

### 5.3.3 QUALITATIVE RESULTS

Qualitative Results are shown in Figure. 5. For KTH results in Figure. 5, we plot a randomly selected sample for all methods. As we can see, SAVP and SVG-LP output average or blurry images after 20-30 frames, and our method is able to switch between action classes (for diverse sample using $\mathcal{GP}$ trigger). In **supplementary**, we show results on KTH with more than 100 sampled frames and best matching samples for baselines and ours. For the BAIR dataset, we show the best LPIPS results for all approaches; where we can see that our method generates samples much closer to the ground-truth. We also included a random sample with a fixed trigger at $15^{\text{th}}$ frame, where we can see a change in the action. For the Human3.6M dataset (after digital zoom), we can see that our best LPIPS sample matches the ground-truth person's pose closely as opposed to SVG-LP, demonstrating the effectiveness of our approach. Similar results are observed for the UCF example. Note that due to manuscript length constraints, we have provided more qualitative results in the **supplementary**.

## 6 CONCLUSION

We propose a method for diverse future video generation. We model the diversity in the potential future states using a $\mathcal{GP}$, which maintains priors on future states given the past and a probability distribution over possible futures given a particular sample. Since this distribution changes with more evidence, we leverage its variance to estimate when to generate from an on-going action and when to switch to a new and diverse future state. We achieve state-of-the-art results for both reconstruction quality and diversity of generated sequences.

**Acknowledgements.** This work was partially funded by independent grants from Facebook AI, Office of Naval Research (N000141612713), and Defense Advanced Research Projects Agency (DARPA) SAIL-ON program (W911NF2020009). We also thank Harsh Shrivastava, Pulkit Kumar, Max Ehrlich, and Pallabi Ghosh for providing feedback on the manuscript.

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

# A  APPENDIX

## A.1  ABLATION STUDIES FOR TEMPORAL DYNAMICS ENCODER

We perform ablation studies on our model by trying different variants of recurrent modules for our temporal dynamics encoder networks. These models are: **DVG-RNN**, our model with an RNN dynamics encoder; **DVG-GRU**, with an RNN dynamics encoder with GRU units.

Figure. A.1 shows ablation analysis for different variants of our approach. On the KTH dataset, different dynamics models (RNN, GRU, LSTM) all perform the same. On the BAIR dataset, RNN perform poorly and LSTM performs the best among the three. On Human3.6M dataset RNN performs higher than our LSTM and GRU models. On the FVD metric in Table A.1, all variants of our approach perform better than all baselines. In approaches, GRU dynamics model performs better on KTH and LSTM performs better on Human3.6M and BAIR dataset.

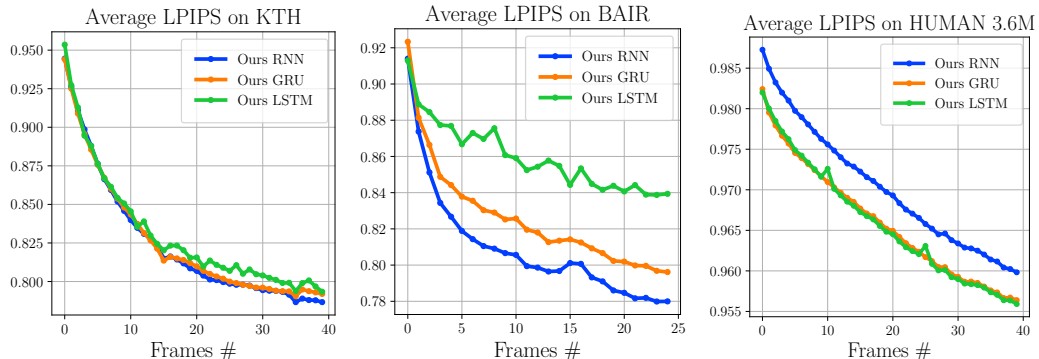

Figure A.1: **Ablation results** on KTH, Human3.6M and BAIR dataset using variants of temporal dynamics model in our method. We report best LPIPS metric. All methods use the best matching sample out of 100 random samples. We used fixed trigger to keep trigger point for each sample the same. On KTH, all temporal dynamics models have similar performance; and on BAIR, our LSTM model have best performance.

Table A.1: **Quantitative results** on KTH, BAIR, Human3.6M datasets. For the **FVD Score**, all the ablation methods use the best matching sample out of 100 random samples and lower numbers are better. For the **Diversity Score**, we compute the score across 50 generated samples, for 500 starting sequences, and higher numbers are better.

| Model | Dynamics | Trigger | FVD Score (↓) | | | Diversity Score (↑) (frames: [10,25]) | | Diversity Score (↑) (frames: [25,40]) | |
|---|---|---|---|---|---|---|---|---|---|
| | | | KTH | BAIR | Human3.6M | KTH | Human3.6M | KTH | Human3.6M |
| DVG [**ours**] | LSTM | @15,35 | 65.69 | **123.08** | **479.43** | 48.30 | **9.3** | **46.20** | 9.0 |
| DVG [**ours**] | GRU | @15,35 | **64.89** | 124.38 | 485.96 | **48.53** | 8.5 | 44.23 | **9.1** |
| DVG [**ours**] | RNN | @15,35 | 66.84 | 126.07 | 503.64 | 46.60 | 7.6 | 41.50 | 8.2 |

## A.2  ANALYSIS: CHANGES IN ACTION AFTER GP TRIGGERING

KTH action dataset comprises of 6 action classes: walking, running, jogging, waving, clapping, and boxing. On an abstract level, we can cluster these actions into two categories moving actions and still actions. From the Figure. A.2, it is interesting to observe that our GP triggering model captures the future trajectories of the videos and clusters them into these two categories moving actions and still actions. Common action switches that are to be expected can be observed from the Figure. A.2; for example, walk and jog, wave and clap interchange frequently after triggering. Still actions seldom change to moving actions.

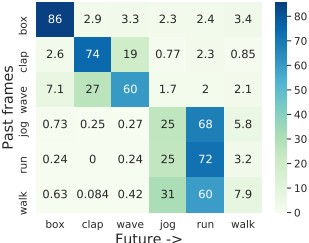

Figure A.2: **Changes in action from past frames to future frames** on KTH dataset. Total of 25,000 generated videos were used to calculate percentage change shown in the above figure.

## A.3 SSIM and PSNR Results

We evaluated our generated video sequences using the tradition metrics like structural similarity index (SSIM) and peak signal-to-noise ratio (PSNR) for comparison with previous baselines which reported these metrics. We trained all models on $64 \times 64$-size frames from the KTH, Human3.6M, and BAIR datasets. We used the standard training practice of using 5 frames as context (or past) and the model have to predict the next 10 frames. For all methods, SSIM and PSNR is computed by drawing 100 samples from the model for each test sequence and picking the best score with respect to the ground truth. We emphasize that these results are only for completeness and we hope that the community will stop relying on such reconstruction metrics for video prediction.

Results are reported in Figure. A.3 represent the evaluation plots for traditional metrics on KTH, BAIR, and Human3.6M dataset. We follow the experimental setups from the baseline papers.

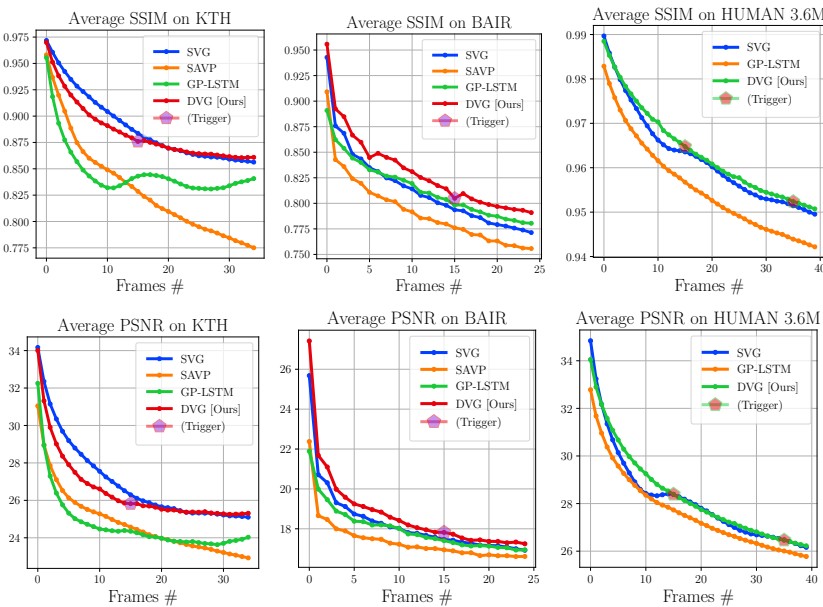

Figure A.3: **Quantitative results** on KTH, BAIR and Human3.6M dataset. We report average SSIM and PSNR metrics. All methods use the best matching sample out of 100 random samples. We used fixed trigger to keep trigger point for each sample the same.

## A.4 Qualitative Results

It can be observed from Figure. A.4 that after 15th frame SVG-LP is stuck in the same pose while after 35th frame SAVP starts distorting the human. However, our method (DVG) consistently generates frames that are diverse and distortion free for longer period of time. Similarly, in Figure. A.9 it can be observed that after 30th frame SVG-LP and SAVP start generating subpar frames while our method is able to generate visually acceptable sequences for longer term. Few additional qualitative results on the BAIR dataset are provided in Figures. A.5-A.6, and on the Human3.6M dataset in Figures. A.7-A.8.

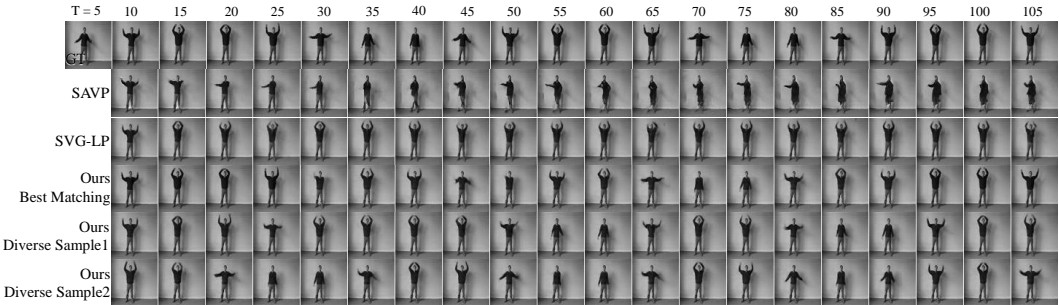

Figure A.4: **KTH dataset:** Qualitative comparison of the generated video sequences (every 5th frame shown). First row is the ground-truth video (with last frame of the provided 5 frames is shown)

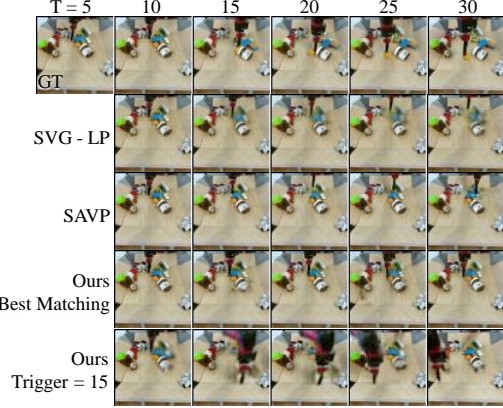

Figure A.5: **Qualitative results** on BAIR dataset. We show the best LPIPS samples out of 100 samples for all methods.

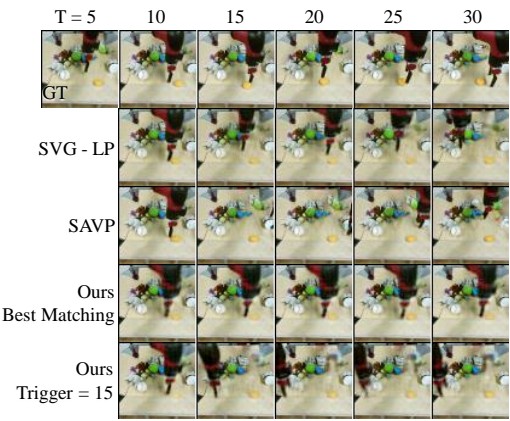

Figure A.6: **Qualitative results** on BAIR dataset. We show the best LPIPS samples out of 100 samples for all methods.

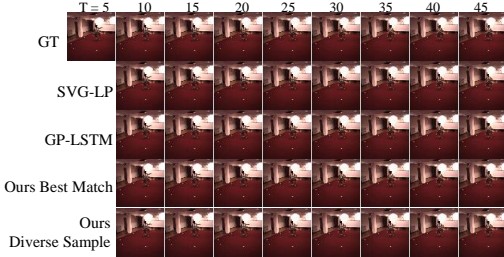

Figure A.7: **Human3.6M dataset:** Qualitative comparison of the generated video sequences (every 5th frame shown). First row is the ground-truth video (with last frame of the provided 5 frames is shown)

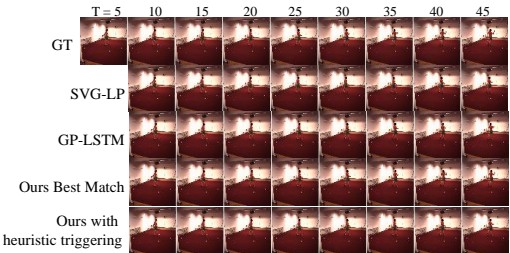

Figure A.8: **Human3.6M dataset:** Qualitative comparison of the generated video sequences (every 5th frame shown). First row is the ground-truth video (with last frame of the provided 5 frames is shown)

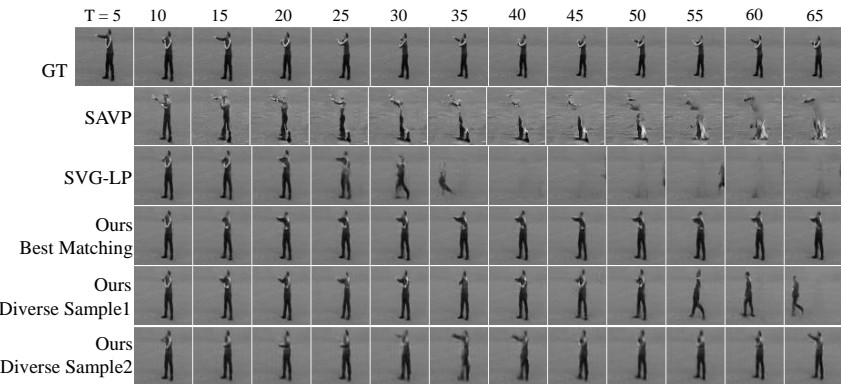

Figure A.9: **KTH dataset:** Qualitative comparison of the generated video sequences (every 5th frame shown). First row is the ground-truth video (with last frame of the provided 5 frames is shown)

## A.5 GAUSSIAN LAYER SPECIFICS

As mentioned in the paper, GPytorch was used for our GP layer implementation. We utilized a large-scale variational GP implementation of GPytorch for our multi-dimensional GP regression problem of learning to predict the variance over the future frames in the latent space. For variational GP implementation, 40 inducing points were randomly initialized and learned during the training of GP. We used a RBF kernel along with gaussian likelihood for our GP layer. For optimization of our GPLayer, we employed stochastic optimization technique (Adam optimizer) to minimize the variational ELBO for a GP.

## A.6 I3D NETWORK ARCHITECTURE FOR ACTION CLASSIFIER

For our diversity metric mentioned in §5, we utilized the standard kinetics-pretrained I3D action recognition classifier. The input to the action classifier is a 15 frames clip and each frame has a size of $64 \times 64$. The action classifier attains accuracy close to 100% for KTH dataset and is above 90% accuracy for human3.6m dataset.

