# OpenReview forum: "Diverse Video Generation using a Gaussian Process Trigger"
_ICLR.cc/2021/Conference — ICLR 2021 Poster_

### Official Review · AnonReviewer1 · 2020-10-20
**GP for Video Generation**

**Rating:** 6
**Confidence:** 4

**Review:**

In this work, the authors propose to apply Gaussian Processes to generate future video frames with high diversity. Specifically, they use variance of GP prediction as a trigger to control when we should switch to a new action sequence.

Strength

1 The paper is written well, and the organization is OK

2 The idea of using GP for video generation sounds interesting

Weakness

1 The way of using GP is kind of straightforward and naive. In the GP community, dynamical modeling has been widely investigated, from the start of Gaussian Process Dynamical Model in NIPs 2005.

2 I do not quite get the modules of LSTM Frame Generation and GP Frame Generation in Eq (4). Where are these modules in Fig.3 ? The D in the Stage 3? Using GP to generate Images? Does it make sense? GP is more suitable to work in the latent space, is it?

3 The datasets are not quite representative, due to the simple and experimental scenarios. Moreover, the proposed method is like a fundamental work. But is it useful for high-level research topics, e.g.,  large-scale action recognition, video caption, etc?

---

> ### Author Response · Authors · 2020-11-12
> **Concerns addressed; Figure updated in the paper for clarification**
>
> We thank R1 for taking the time to go through the paper thoroughly and understanding the gist of the paper, especially for acknowledging the elegance of our idea.
>
> **Using GP is straightforward**
> We thank the reviewer for referring us to the previous work in GP literature, which closely resembles our intuition. We agree that modeling dynamical systems with GP is a vibrant community, and GP has been used in other communities (e.g., planning for robotics). However, to the best of our knowledge, no prior work has explored the use of GP in modeling the multimodal nature of video generation/prediction. Our work demonstrates that a simple GP model can beat the current state-of-the-art by a considerable margin in terms of both diversity and fidelity. We strongly believe that GP-based approaches are a better way to capture the multimodal nature of the video generation task and hope that the future line of work investigates such methods further. Therefore, we believe that our work deserves a wider audience of this conference.
>
> **Architecture clarification**
> LSTM frame generation and GP frame generation are not entirely different modules but are used as additional loss terms to train the decoder and encoder network. In Figure 3, they are depicted separately under “Training Dynamics Encoders” for clarity. The joint framework is shown for inference in Figure 3, which depicts how they are used together. Since the decoder network maps the latent space to the image space, it is trained using three different frame generation loss functions given by the frame auto-encoder loss ($z_t$ ~ frame encoder), the LSTM frame generation loss ($\hat{z}_t$ ~ LSTM temporal dynamics encoder), and the GP frame generation loss ($\tilde{z}_t$~ GP temporal dynamics encoder). As for the concern, yes, we utilize the GP only in the latent space.
>
> **Datasets and High-level Research topics**
> The datasets utilized for the experiments are standard in this community and were used to make a fair comparison with the baselines in terms of quantitative evaluations. We also demonstrate qualitative evaluation on a more representative dataset like UCF-101, which has videos in the wild setting.
>
> We thank the reviewer for realizing that the proposed approach is a fundamental contribution that can be used to better understand other research areas in videos (e.g., video recognition). Though we agree with the comment, we emphasize that this paper only focuses on the task of diverse video generation. Adapting our approach to build better models in other applications is indeed an area of future research.
>
> **Updates to the paper:**
> * Added GPDM line of work in the related work section.
> * Updated figure 3

---

### Official Review · AnonReviewer2 · 2020-10-28
**Review for Diverse Video Generation using a Gaussian Process Trigger**

**Rating:** 6
**Confidence:** 3

**Review:**

### SUMMARY

The authors propose to use a Gaussian Process (GP) to model the uncertainty of future frames in a video prediction setup. In particular, they employ a GP to model the uncertainty of the next step latent in a latent variable model. This allows them to use the GP variance to decide when to change an "action sequence", corresponding to a deterministic dynamics function implemented using an LSTM.

### STRENGTHS AND WEAKNESSES

[+] Empirical results

[+] Well-motivated model

[+] Clear presentation

[-] Experimental section could be improved (missing baselines in some tables, results seem to differ from those in the literature)

### DETAILED COMMENTS

The paper proposes a novel approach for video prediction. Following the standard latent variable model setup used by many VAE-based video prediction models, the authors propose to use a GP to model the uncertainty in the latent space while also learning a deterministic dynamics model (LSTM) on this latent space. Then the GP is used to decide when a future frame has high uncertainty, and in those cases multiple latents can be sampled from the GP. In general the paper is clear and well-written.

The experimental section could be improved. In particular, more details about how the comparison to some baselines was made would be appreciated. For example,  the results for the VRNN model in Figure 4 and 5 do not follow the results in the literature, where it outperforms SVG and SAVP, and its unclear whether its due to an architectural change, suboptimal hyperparameters, or a different reimplementation. Further this model is missing from some other comparisons such as Table 1. For SAVP the results for Figure 4 seem much worse than those reported in the original paper. On the other hand, the authors did some ablation experiments and included different metrics to analyze the performance of their method.

### SCORE
I vote for accepting the paper. The model formulation is clear, well-motivated and novel. The results are positive and overall it seems like a valid alternative to current approaches that will be of interest to the video prediction community. I would encourage the authors to provide a more thorough experimental section.

### POST-REBUTTAL UPDATE
After reading the other reviews and the authors' rebuttal, I stand by my rating of 6.

---

> ### Author Response · Authors · 2020-11-12
> **Addressed Concerns about experimental setup; Added additional results in the revised paper**
>
> We want to thank the reviewer for all the valuable comments, especially for finding the paper well-motivated, good in terms of empirical results and presentation.
> We would like to address the concerns regarding the comparison with baselines. Wherever available, we used either the official implementation or the pre-trained models for the baselines uploaded by their respective authors. We have added this clarification to the revised manuscript.
>
> **Results of VRNN**
> We used the official implementation of VRNN to train the baseline models since pre-trained models were not released. We followed all the best practices and ensured that the results (qualitative and quantitative) match or are better than the original paper. Our LPIPS results on the BAIR dataset (Figure 4-center) are in-line with Figure 6 of [VRNN], where the performance of SAVP, SVG, and VRNN are close, and VRNN is slightly better. We have also updated the two missing numbers for VRNN in Table 1 in the revised manuscript.
>
> **SAVP results**
> We used the pre-trained model released by the authors for the SAVP baseline. For both the KTH and BAIR datasets, we followed the testing protocol from SVG’s official implementation (for each test video, sample 100 sequences, and evaluate the best matching sequence). The results from the original paper [SAVP] (Figure 8, row-1, col-3) show that SAVP is much worse compared to SVG (*with fixed prior*), especially for later frames. In our paper (Figure 4, KTH), we observe that SAVP is worse than SVG (*with learned prior*). A more considerable margin is explained by a stronger SVG model where the prior is learned. The magnitude of results are similar across Figure 4 (our paper) and Figure 8 ([SAVP]).
>
> **Updates to the paper:**
> * Added the VRNN diversity scores on the Human3.6M dataset to Table 1.
> * Added clarifications stating that we used official implementations/pre-trained models before the start of descriptions of the baselines.
>
>
> [SAVP] Lee, Zhang, Ebert, Abbeel, Finn, Levine. ‘Stochastic Adversarial Video Prediction’
>
> [VRNN] Castrejon, Ballas, Courville. ‘Improved Conditional VRNNs for Video Prediction’

---

### Official Review · AnonReviewer3 · 2020-10-28
**Official Blind Review #3**

**Rating:** 6
**Confidence:** 3

**Review:**

Summary:
This paper proposes a future frame prediction framework where the video generation can transition between different actions using a Gaussian process trigger. The framework consists of three components: an encoder which encodes the frame to a latent code, an LSTM which predicts the next latent code given the current one, and a Gaussian process which samples a new latent code. The framework can decide whether to switch to the next action by adopting the new latent code, depending on the number of frames passed or the variance of Gaussian.

Strengths:
The paper is easy to follow overall. The usage of Gaussian process to trigger the transition to the next action is reasonable and intuitive. Quantitative evaluations show that the method outperforms existing works for both reconstruction and output diversity for various datasets.

Weaknesses and comments:
There are quite a few typos in the writing, especially toward the latter part of the paper. I’d encourage the authors to do a thorough check to ensure the paper is typo-free.
It seems switching actions at some fixed number of frames beats using the Gaussian variance for FVD, which is quite surprising. Can the authors provide some insights? Is it due to some inherent nature of FVD, or there’s still some room for improvement for the choosing criteria?
How important is the heuristic of changing states when using GP? Currently it is triggered when the variance is larger than two standard deviations. How will it affect the performance if a different threshold is used?
There’s a mistake in Table 1. The diversity score for DVG@15,35 is the best for KTH frames [10,25] (48.30), but DVG GP is bolded (47.71). This might also be an interesting point to discuss about why fixed number of frames performs better than GP.

---

> ### Author Response · Authors · 2020-11-12
> **Clarifications on results; paper revised**
>
> We want to thank the reviewer for thoughtful comments; especially for acknowledging the intuition behind using a GP based approach to solve an important problem of diverse video generation and the fact that sufficient empirical evidence was put forth to back our claims.  We are glad to incorporate suggestions put forth here.
>
> **GP Triggering (FVD)**
> For finding the FVD scores, as mentioned in our paper, we take the best matching sample to the ground-truth. When we use GP variance as a trigger, the chances of getting the best matching are lower as there are multiple plausible (diverse) futures that can be picked after the trigger. In theory, we should be able to achieve the best ground-truth matching sample, if we are allowed to sample infinite times; however, for our evaluations, we keep the upper limit to 100 random samples per starting sequence. Hence, we observe better scores for the deterministic trigger DVG@15,35
>
> **Insights for GP Heuristics**
> While running our experiments, we found that the range of variance of the learned GP is different for different starting sequences and actions. For example, sequences from box action tend to have less spread in the variance values, whereas those from walk action have more spread. This makes picking a fixed threshold challenging, and hence our design choice of using greater than two standard deviations. Changing this to [1.5, 2.5]x standard deviations gives similar results. Lower than this range triggers changes in sequence before the action finishes, and higher values tend not to trigger diverse samples. Therefore, we opted for a simple design choice of using two-sigma, which gives us a sequence-dependent threshold.
>
> **Diversity Score for DVG@[15,35]**
> We appreciate the thoroughness of the reviewer in raising this point -- `fixed number of frames performs better than GP triggering during the evaluation of the first clip of the generated sequence. We observe this phenomenon because our two-sigma heuristic makes conservative decisions on when to trigger the change from an ongoing action sequence. Hence, during the evaluation of the first clips, the number of videos triggered with the GP variance trigger is less than the number of videos triggered by the deterministic trigger as the deterministic trigger forces every video to trigger at the 15th frame. However, this is not the case when we perform our second clip evaluation as the number of GP triggers increases as the sequence progresses.
>
> Lastly, we fixed the typos in our revised manuscript, as suggested. We again want to thank the reviewer for providing us with valuable feedback.

---

### Author Response · Authors · 2020-11-22
**New Revision Uploaded**

Dear AC and Reviewers,

Thank you for your valuable suggestions. We have uploaded a new version of the paper with all of the changes we discuss in our individual responses below. In particular, we added new numbers in Table 1 of our updated paper. We also took the liberty to improve some of the figures based on the feedback. We are happy to answer any more of your concerns during this discussion period.

Thank you,

The Authors

---

### Decision · Program_Chairs · 2021-01-07
**Final Decision**

**Decision:**

Accept (Poster)

**Comment:**

All three reviewers agree on accepting the paper and think that the proposed approach will be of interest for those working in vdieo prediction.  The authors are asked to include the extra discussion with R3 as part of the paper and include the proposed changes by R2 to provide more thorough experimentation.  The paper is recommended as a poster presentation.